# Differing trabecular bone architecture in dinosaurs and mammals contribute to stiffness and limits on bone strain

Trevor G. Aguirre[1], Aniket Ingrole[2☯], Luca Fuller[2☯], Tim W. Seek[1☯], Anthony R. Fiorillo[3¤‡], Joseph J. W. Sertich[4‡], Seth W. Donahue[2]*

**1** Mechanical Engineering Department, Colorado State University, Fort Collins, Colorado, United States of America, **2** Department of Biomedical Engineering, University of Massachusetts-Amherst, Amherst, Massachusetts, United States of America, **3** Perot Museum of Nature and Science, Dallas, Texas, United States of America, **4** Department of Earth Sciences, Denver Museum of Nature & Science, Denver, Colorado, United States of America

☯ These authors contributed equally to this work.
‡ These authors also contributed equally to this work.
¤ Current address: Huffington Department of Earth Sciences, Southern Methodist University, Dallas, Texas, United States of America
* swdonahue@umass.edu

**Data Availability Statement:** All relevant data are within the manuscript and its Supporting Information files. CT scans of extant species of this study are available online at: https://doi.org/10.

## Abstract

The largest dinosaurs were enormous animals whose body mass placed massive gravitational loads on their skeleton. Previous studies investigated dinosaurian bone strength and biomechanics, but the relationships between dinosaurian trabecular bone architecture and mechanical behavior has not been studied. In this study, trabecular bone samples from the distal femur and proximal tibia of dinosaurs ranging in body mass from 23–8,000 kg were investigated. The trabecular architecture was quantified from micro-computed tomography scans and allometric scaling relationships were used to determine how the trabecular bone architectural indices changed with body mass. Trabecular bone mechanical behavior was investigated by finite element modeling. It was found that dinosaurian trabecular bone volume fraction is positively correlated with body mass similar to what is observed for extant mammalian species, while trabecular spacing, number, and connectivity density in dinosaurs is negatively correlated with body mass, exhibiting opposite behavior from extant mammals. Furthermore, it was found that trabecular bone apparent modulus is positively correlated with body mass in dinosaurian species, while no correlation was observed for mammalian species. Additionally, trabecular bone tensile and compressive principal strains were not correlated with body mass in mammalian or dinosaurian species. Trabecular bone apparent modulus was positively correlated with trabecular spacing in mammals and positively correlated with connectivity density in dinosaurs, but these differential architectural effects on trabecular bone apparent modulus limit average trabecular bone tissue strains to below 3,000 microstrain for estimated high levels of physiological loading in both mammals and dinosaurs.

6084/m9.figshare.7257179.v1 Architectural index data for extant species are available in the Supplemental Information for references 8 and 9. CT scans of the extinct species can be obtained by contacting the relevant researchers in the acknowledgements section.

**Funding:** Funding was provided by the National Science Foundation Office of Polar Programs (OPP 0424594), as well as the National Geographic Society (W221-12) for the collection of Alaska Edmontosaurus materials used here. And, the Arctic Management Unit of the Bureau of Land Management provided administrative support. The specimens discussed here were collected under BLM permit number AA–86367. Travel funding for Mammuthus columbi sample collection was provided by the George C. Frison Institute of Archaeology and Anthropology. The funders had no role in study design, data collection and analysis, decision to publish, or preparation of the manuscript.

**Competing interests:** The authors have declared that no competing interests exist.

## Introduction

Terrestrial dinosaurs were massive animals that placed exceptional mechanical demands on their bones, but it is unknown how trabecular bone architecture helped meet those demands. Bones need to be sufficiently strong and robust to resist fracture during habitual physical activity. However, cellular maintenance and transport of bone during locomotion is metabolically expensive. Thus, bony architecture must achieve mechanical competence while maintaining low weight. If an individual bone was so large that the mechanical strains were very low during routine activities such as walking and running, the animal would expend unnecessary energy to move a large heavy skeleton. However, if mechanical loading becomes too large on bones, the risk of failure increases [1, 2]. The physiological process of bone remodeling helps achieve a balance between bone weight and mechanical competence, repairs and limits the accumulation of fatigue damage [3], and adapts to limit strains [4]. Bone has adapted at the architectural and tissue levels to its mechanical environments [5], including some highly specialized bone structures to meet exceptional mechanical demands [6]. For example, the horncore of bighorn sheep (*Ovis canadensis*) has a unique porous bone architecture that absorbs energy and reduces brain cavity accelerations during impact [7]. Trabecular bone is a highly porous architectural foam that provides lightweight mechanical competence. It has been shown that trabecular bone architectural indices (trabecular connectivity density, thickness, number, and spacing) scale allometrically with body mass in animals ranging in size from mouse to elephant [8, 9]. From previous studies, it is noteworthy that trabecular bone volume fraction does not scale with body mass in mammalian and avian long bones. This is surprising because it is well documented that the mechanical properties of trabecular bone positively correlate with volume fraction [10]. From an evolutionary perspective, it is possible that trabecular bone volume fraction does not increase with body mass because the bones would become too heavy and therefore too metabolically expensive to maintain and transport.

Trabecular bone architecture may be adapted and organized in a way that provides stiffer trabecular bone material behavior at the continuum level in more massive animals. It has been shown that more massive animals have improved mechanical properties such as higher fatigue strength in cortical bone material [11]. Whole bones are known to become more robust with increasing body mass [12–15]. Well-developed trabecular bone architecture in more massive animals would help minimize the total amount of bone material needed to build a more robust whole bone and therefore help minimize metabolic energy costs of cellular maintenance and transportation. Trabecular architecture is related to mechanical performance. For example, decreased trabecular thickness has been shown to cause a 2-5x reduction in strength [16] and increased connectivity density is associated with increased strength [17, 18]. Finite element models have been utilized to investigate the biomechanics of dinosaur trabecular bone [19–21]. It was found that the trabecular bone architecture in plesiomorphic theropods more closely resembles the trabecular architecture of modern humans than that of extant avian species, implying similar biomechanics to humans. These conclusions were made based on the oblique nature of the trabeculae in the proximal femoral metaphysis being like that of humans. Previous research on dinosaur trabecular bone has contributed to our understanding of locomotor behavior [19–21]. However, the effects of dinosaur trabecular bone architecture on mechanical performance have not been determined.

Dinosaur trabecular bone experienced extreme mechanical loads due to extreme body mass (up to 47,000 kg) [22, 23]. Because bone remodels to optimize mechanical performance and weight in response to habitual mechanical loading [5, 24], we hypothesized that the trabecular architecture in the long bones of large mass dinosaurs confers a higher specific apparent elastic modulus compared to animals with smaller body mass preventing bone strains from reaching

dangerous levels. Finite element models of trabecular bone were constructed from computed tomography (CT) and micro-computed tomography (μCT) scans of trabecular bone cores from extinct species with body masses up to 9,980 kg. The finite element models were used to assess the effects of body mass on trabecular bone elastic modulus and principal strains. These findings help explain how dinosaur skeletons supported such massive loads. Additionally, dinosaur trabecular bone architectures adapted to extreme mechanical environments could have implications for novel bioinspired engineering applications [25].

## Materials and methods

### Species analyzed

The species used in the finite element models of this study were chosen to cover a wide range of body masses, from 1 to ~10,000 kg, and are listed in Table 1. CT and μCT scans from previous studies were used for some species as indicated in Table 1. For *Mammuthus columbi* and *Edmontosaurus*, new bone samples were obtained and scanned with μCT for the current study. One sample was used for each specimen listed in Table 1. The mammalian species utilized in this study are all quadrupedal whose primary form of locomotion is walking and running. The hadrosaur specimens were largely quadrupedal but capable of bipedal motion when necessary [26–28]. All theropods were bipedal. Furthermore, the hadrosaur and theropods primary form

**Table 1. Species used in this study.**

| Study | Common Name | Specimen Number | Species | Body mass (kg) |
|---|---|---|---|---:|
| [9] | Java Mouse Deer | UMZC H15013 | *Tragulus javanicus* | 1 |
| [19] | Troodontid | MOR 748 | *Troodontidae* | 23 |
| [19] | Caenagnathid | TMP 1986.036.0323 | *Caenagathidae* | 49 |
| [9] | Domestic sheep | RVC sheep2 | *Ovies aries* | 57 |
| [19] | Ornithomimid | TMP 1999.055.0337 | *Ornithomimidae* | 100 |
| [19] | Therizinosaur | UMNH VP 12360 | *Falcarius Utahensis* | 128 |
| [9] | Siberian Tiger | RVC tiger_2 | *Panthera tigris* | 130 |
| Current | Hadrosaur | [a]DMNH 22386 | *Edmontosaurus annectens* | 420 |
| Current | Hadrosaur | [a]DMNH 22231 | *Edmontosaurus annectens* | 420 |
| Current | Hadrosaur | [a]DMNH 22235 | *Edmontosaurus annectens* | 420 |
| Current | Hadrosaur | [a]DMNH 2012 25–57 | *Edmontosaurus annectens* | 420 |
| Current | Hadrosaur | [a]DMNH 22228 | *Edmontosaurus annectens* | 420 |
| Current | Hadrosaur | [a]DMNH 22242 | *Edmontosaurus annectens* | 420 |
| [9] | White Rhinoceros | RVC french_rhino | *Ceratotherium simum* | 3,000 |
| [9] | Asian Elephant | RVC gita | *Elephas maximas* | 3,400 |
| Current | Hadrosaur | [b]DMNH 44398 | *Edmontosaurus regalis* | 7,936 |
| Current | Hadrosaur | [b]DMNH 42169 | *Edmontosaurus regalis* | 7,936 |
| Current | Mammoth | 48WA322-9 | *Mammuthus columbi* | 9,980 |

University Museum of Zoology Cambridge (UMZC), Cambridge, United Kingdom, Europe. Royal Veterinary College (RVC), London, United Kingdom, Europe.

Museum of the Rockies (MOR), Bozeman, Montana, United States of America.

Royal Tyrrell Museum of Palaeontology (TMP), Drumheller, Alberta, Canada.

Natural History Museum of Utah (UMNH), Salt Lake City, Utah, United States of America. Perot Museum of Nature and Science ([a]DMNH), Dallas, Texas, United States of America.

Denver Museum of Nature & Science ([b]DMNH), Denver, Colorado, United States of America.

University of Wyoming Archaeological Repository (UWAR), Laramie, Wyoming, United States of America. 48WA is the archaeological site identification code per the Smithsonian trinomial system.

of locomotion is walking and running [29, 30]. Sex is unknown for all species, the mammals were skeletally mature [9], and age or skeletal maturity is unknown for all dinosaurian species as this is difficult to determine from the fossil record [31].

The body mass estimations for the extinct species of this study are as follows: *Edmontosaurus regalis* 7,936 kg [29], *Edmontosaurus annectens* 420 kg [29], *Troodontid* 23kg [32], Caenagnathid 49kg [33], *Falcarius utahensis* 128 kg [34], Ornithomimid 100 kg [35], and *Mammuthus columbi* 9,980kg [36]. For the *Mammuthus columbi* the body mass estimation is for the specific specimen used in this study. For the other species, the body masses were obtained from the published estimates shown above and were assumed to be the same for all specimens of a given species.

## Computed tomography scanning

For the species in this study, trabecular bone samples from the medial portion of either the proximal tibia or distal femur were analyzed based on availability (Fig 1). These locations were selected because of similarities in the trabecular bone architectural indices in these two regions [37]. Archival μCT scans of trabecular bone from the lateral femoral condyles were accessed via a public database [38]. High-resolution CT scans of fossilized dinosaur limbs [19–21] were provided by Dr. Peter Bishop at the Royal Veterinary College in the United Kingdom. Sections

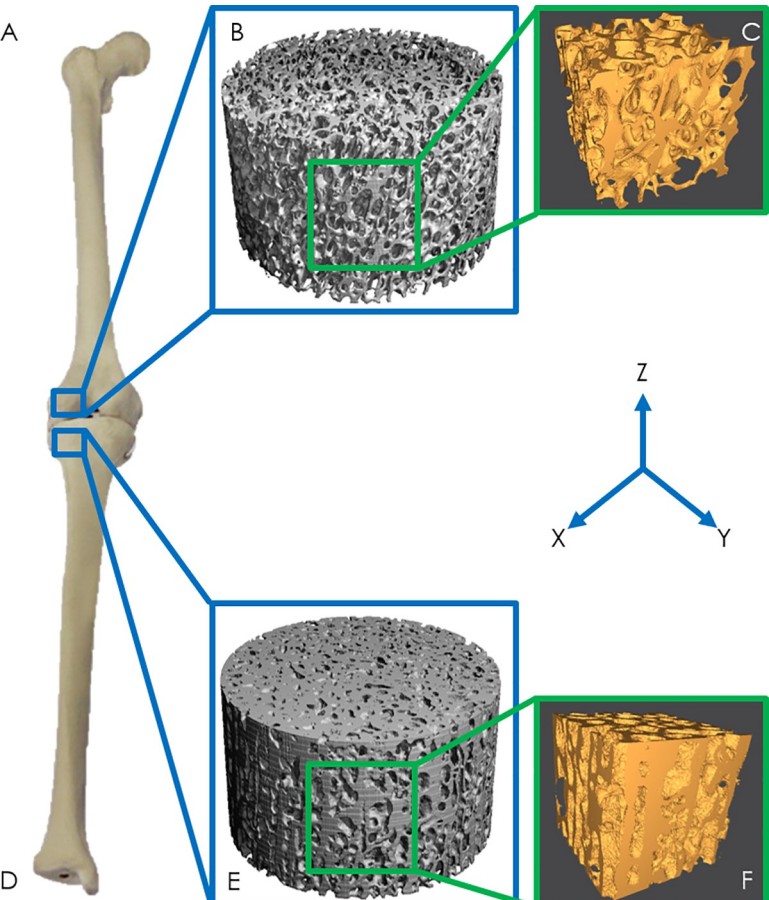

**Fig 1. A)** Femoral core location, **B, E)** μCT scans of trabecular cores, **C, F)** finite element models of trabecular bone, **D)** Tibial core location.

of trabecular bone were virtually cropped from the lateral femoral condyle in the CT scans. New cylindrical cores of trabecular bone were collected from several hadrosaur specimens. Two hadrosaur (*Edmontosaurus annectens*) tibiae were provided by the Denver Museum of Nature & Science. Six hadrosaur (*Edmontosaurus* sp.) tibiae were provided by the Perot Museum of Nature and Science. Additionally, a femoral core was collected from a Columbian mammoth (*Mammuthus columbi*) in the University of Wyoming Archaeological Repository fossil collection. Fig 1 displays the anatomical locations from which cores for this study and from previous studies [9, 19, 38] were obtained. The new trabecular cores collected for this study were harvested using a diamond sintered coring bit and were 8 mm in diameter and 50–75 mm long. During drilling, water was pumped through the center of the coring bit to cool the sample/bit and flush out debris.

The trabecular bone volume fraction (BV/TV), trabecular thickness (Tb.Th), trabecular spacing (Tb.Sp), and connectivity density (Conn.D) for each CT scan [9, 19] were measured using BoneJ [39] and the trabecular number (Tb.N) was computed using the methods in [40]. The new trabecular cores were scanned with a SCANCO micro-computed tomography machine (SCANCO μCT 80) at high resolution, 8W, and 70 kV peak excitation voltage to produce 10-micron voxels. To prevent image distortion, fossilized trabecular bone cores were scanned through a copper filter [41].

## Finite element model generation

The CT and μCT DICOM files were binarized with Seg3D to separate the bony material from the marrow space. Finite element models were generated by cropping a cube from the center of the cylindrical scan volume (Fig 1C & 1F). This location was chosen so that peripheral damage from coring was not included in the finite element models. Bulk dimensions of the finite element models varied due to differences in the available μCT scan regions of intact bone (e.g., some *Edmontosaurus* and the *Mammuthus columbi* samples had irregular geometries due to the coring process). However, all finite element models had the dimensions required to treat trabecular bone as a continuum, which is 5–10 trabecular spacings [42]. Sample image files were exported in the ASCII STL file format for further file preparation. MeshMixer was used to create a solid volume from the surface model exported from Seg3D and to repair any errors during surface triangulation. The files were then meshed in ICEM CFD to generate a linear tetrahedral element mesh and finite element models were generated using ABAQUS.

## Finite element modeling

Quasi-static compression simulations were performed on each finite element model. The solid bone material within the finite element models was assigned an elastic modulus of 15 GPa [43, 44], Poisson's ratio of 0.3 [43, 44], and modeled as a linear elastic material. We were interested in the strain distributions (as an indicator of failure risk) in the trabecular bone during high levels of estimated physiological loading. Therefore, compressive loading was simulated through the application of an apparent level strain equal to one half of the trabecular bone compressive yield strain (4,150 microstrain). The compressive yield strain (8,300 +/- 100 microstrain) of trabecular bone was used because it is remarkably similar across a large range of relative densities and animals with a large range of body masses [45, 46]. Through the application of equal apparent strain to each trabecular bone cube, the effect of the trabecular architecture on the apparent modulus and principal strains could be directly assessed [9]. The apparent level strain was applied using displacement magnitudes based on the height of each trabecular bone cube. These displacements were applied to the top nodes of each finite element model in the direction of the bone long axis (z-direction in Fig 1) by using a roller-type

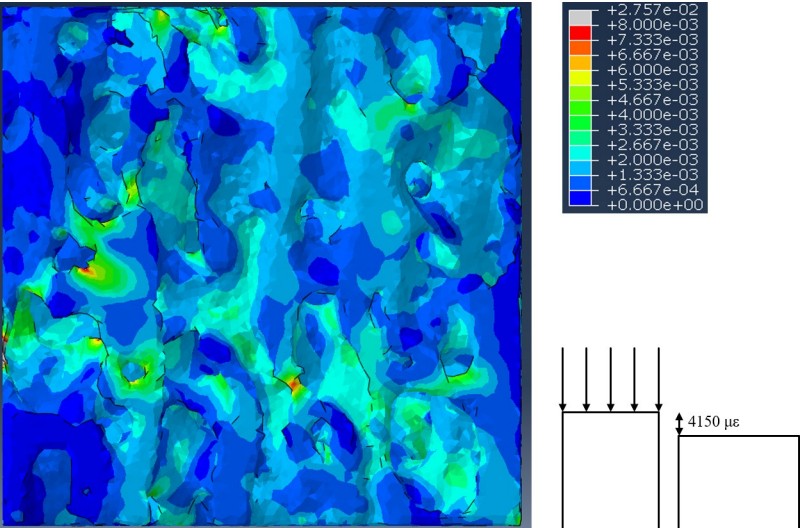

**Fig 2. Example finite element model after compression to a macroscopic strain of 4,150 microstrain (με).** The color gradient corresponds to the max principal strain in each element.

boundary condition. The nodes on the bottom surface of the finite element models were constrained in the z-direction only with frictionless contacts. An example finite element model is shown in Fig 2. Finite element models were generated for all species listed in Table 1.

To determine the optimal mesh for the finite element models a mesh (numerical) convergence study [47, 48] was performed. For this study, five unique mesh densities, ranging from 50,797 to 1,019,808 elements per cubic millimeter, were created for the trabecular bone specimen with the smallest average trabecular thickness and subjected to a strain of 4,150 microstrain. To determine whether the mesh had converged, the change in strain energy between each mesh was analyzed and compared to the finest mesh as a percent difference using Eq 1.

$$\Delta = \frac{X_N - X_i}{X_N} 100\% \tag{1}$$

Where $\Delta$ is the percent difference and X is strain energy. $X_N$ is strain energy for the finest mesh in the mesh convergence study and $X_i$ is the strain energy for the other meshes used in the study. Convergence was achieved at a mesh density of 435,725 elements per cubic millimeter, which had a 3% difference from the finest mesh density of 1,019,808 elements per cubic millimeter.

The effect of the trabecular bone architecture on the apparent elastic modulus ($E_{App}$) was determined by dividing the peak apparent stress divided by the applied displacement for each finite element model. The peak apparent stress was computed by dividing the peak load by the specimen area, which computed from the bulk dimensions of each finite element model cube. To account for BV/TV differences between each cube the specific apparent elastic modulus ($E_{App\ Spec}$) was computed by dividing the apparent elastic modulus by the product of the bone volume fraction and a trabecular bone tissue density of 1.874 g/cm$^3$ [49]. This value was used and assumed to be the same for all samples because the bone tissue density of the fossilized samples could not be accurately measured due to the fossilization process. By keeping the density constant across all samples, we were able to assess the effects of trabecular architecture on the modulus and principal strains.

In addition to apparent stiffness, we were interested in how trabecular architecture impacted the failure risk in each sample. Traditional engineering failure theories such as the distortion energy theory, maximum normal stress, maximum shear stress, and maximum strain energy density do not accurately predict failure of bone tissue due to material anisotropy [50–52]. The Tsai-Wu failure theory also does not work because the planar failure envelopes were found to be uncoupled from each other during biaxial [53] and triaxial [54] loading of bovine trabecular bone. The modified super ellipsoid failure theory improves on previous approaches, but is anatomic site and patient-specific [45]. Therefore, to assess the likelihood of failure of the samples in this study, trabecular principal strains were analyzed directly. The normal and shear strain components were collected from element centroids for every element in each finite element model using a custom Python script. Data were collected from element centroids because this is the location of the Gauss (integration) point for a linear tetrahedral element [55]. A custom MATLAB script was used to compute the principal strains for each model by computing the eigenvalues of the 3D strain tensor [56–58]. The average tensile and compressive principal strains (of all elements) were computed for each finite element model. Additionally, as an indicator of failure risk, the average tensile and compressive principal strains were computed for each finite element model only considering elements that had strain values that exceeded the tensile ($\varepsilon_y$ = 4,100 microstrain) or compressive ($\varepsilon_y$ = -8,300 microstrain) yield strains of human trabecular bone. We refer to these as the largest tensile and compressive principal strains. The yield strains for human trabecular bone were used because the yield strains are narrowly distributed [46, 59, 60]. These four strain parameters were regressed against body mass.

## Allometric scaling relationships

To determine how the trabecular bone architectural indices of the specimen used in this study scale with body mass, log-log plots for these properties were created and compared to extant mammalian and avian species. Allometric scaling relationships were created for the trabecular bone architectural indices versus body mass by linearization of the equation $y = a \cdot x^b$ [61] through a base-10 logarithmic transformation such that:

$$log_{10}(y) = log_{10}(a) + b \cdot log_{10}(x) \tag{2}$$

In Eq 2, $log_{10}(y)$ and $log_{10}(x)$ are the logarithmically transformed trabecular bone architectural indices and body mass, respectively, and $log_{10}(a)$ and b are the y-intercept and slope, respectively, from the linear regressions performed on base-10 logarithmically transformed values for the trabecular bone architectural indices and body mass [62]. Architectural indices for the samples in this study (Table 1) were compared to those from the proximal tibia or distal femur of mammalian [8, 9] and avian [9] species.

## Statistical analyses

Linear regressions between trabecular bone architectural indices and body mass were made to determine allometric scaling relationships for mammalian, avian, and dinosaurian species. Pairwise comparisons were made between regression slopes of the mammalian, avian, and dinosaurian species using a Tukey post-hoc test. In the pairwise comparisons, species was used as a categorical predictor with dinosaurian species used as the reference level. Linear stepwise regressions were used to determine if the trabecular bone architectural indices predict the apparent and specific apparent elastic moduli. The candidate independent variables were Tb. Th, Tb.Sp, and Conn.D, and the dependent variables were apparent elastic modulus and specific apparent elastic modulus. Trabecular number and bone volume fraction were excluded

from stepwise regression models to avoid collinearity since both of these parameters are dependent on trabecular thickness and trabecular spacing [40]. Independent variables were rejected if p > 0.1. For the stepwise regressions the mammalian and dinosaurian apparent and specific apparent elastic moduli data from the finite element models were analyzed separately. Similarly, the apparent elastic modulus, specific apparent elastic modulus, and principal strains for the dinosaurian and mammalian species were analyzed separately for linear regressions versus body mass. Pairwise comparisons were made between the regression slopes for data from the finite element models. Linear regressions, pairwise comparisons, and stepwise regressions were computed using Minitab (version 18). Due to the imbalance between the numbers of dinosaurian samples the average values for *Edmontosaurus regalis* and *Edmontosaurus sp.* were used in all regressions. Due to the low number of dinosaur samples we let α = 0.1 to reduce the chance of Type II error [63–65].

## Results

### Allometric scaling of trabecular bone architectural indices

Allometric scaling relationships indicate that for mammals, bone volume fraction, trabecular thickness, and trabecular spacing show positive correlation with body mass, and trabecular number and connectivity density show negative correlation with body mass. For the avian species, the regressions indicate that bone volume fraction and trabecular thickness show positive correlation with body mass, and trabecular number and connectivity density show negative correlations with body mass. For the dinosaurian species, positive correlation (p < 0.09) with body mass is observed for bone volume fraction, trabecular number, and connectivity density and negative correlation with body mass for trabecular spacing. The allometric regression results are shown in Table 2.

### Finite element modeling

The apparent modulus and specific apparent modulus are shown in Figs 3 and 4, respectively. For the dinosaurian species, positive correlation with body mass is observed for apparent

**Table 2. Allometry linear regression results: Slope (b), with 95% confidence intervals (CI), intercept ($log_{10}(a)$), coefficient of determination ($R^2$), and p-values for the regression slopes.**

| Class | | b | -CI | +CI | $log_{10}(a)$ | $R^2$ | p |
|---|---|---|---|---|---|---|---|
| Mammalian | BV/TV (%) | 0.040 | 0.02 | 0.06 | 1.425 | 0.161 | <0.001 |
| | Tb.Th (μm) | 0.156 | 0.14 | 0.18 | 2.020 | 0.726 | <0.001 |
| | Tb.Sp (μm) | 0.106 | 0.09 | 0.12 | 2.545 | 0.583 | <0.001 |
| | Tb.N (mm$^{-1}$) | -0.118 | -0.13 | -0.10 | 0.334 | 0.698 | <0.001 |
| | Conn.D (mm$^{-3}$) | -0.376 | -0.42 | -0.33 | 1.449 | 0.763 | <0.001 |
| Avian | BV/TV (%) | 0.146 | 0.02 | 0.28 | 1.021 | 0.249 | 0.030 |
| | Tb.Th (μm) | 0.238 | 0.17 | 0.31 | 2.125 | 0.761 | <0.001 |
| | Tb.Sp (μm) | 0.069 | -0.11 | 0.24 | 3.209 | 0.039 | 0.416 |
| | Tb.N (mm$^{-1}$) | -0.081 | -0.24 | 0.08 | -0.249 | 0.061 | 0.306 |
| | Conn.D (mm$^{-3}$) | -0.524 | -0.79 | -0.26 | 0.556 | 0.513 | <0.001 |
| Dinosaurian | BV/TV (%) | 0.068 | -0.02 | 0.15 | 1.410 | 0.552 | 0.091 |
| | Tb.Th (μm) | -0.115 | -0.40 | 0.17 | 2.753 | 0.235 | 0.330 |
| | Tb.Sp (μm) | -0.185 | -0.37 | 0.00 | 3.036 | 0.649 | 0.053 |
| | Tb.N (mm$^{-1}$) | 0.170 | -0.04 | 0.38 | -0.241 | 0.549 | 0.092 |
| | Conn.D (mm$^{-3}$) | 0.631 | -0.10 | 1.36 | -0.619 | 0.591 | 0.074 |

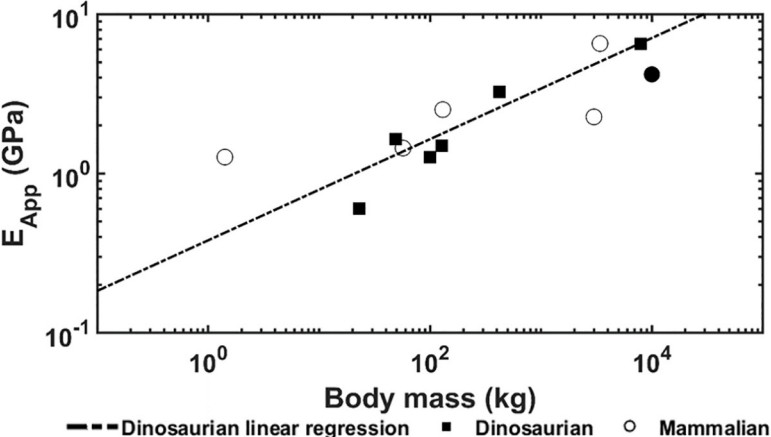

**Fig 3. Apparent elastic modulus versus body mass.** Trabecular bone apparent modulus is positively correlated with body mass in dinosaurs, while for mammalian species no correlation is observed. The solid circle indicates the mammoth.

(p = 0.007, $R^2$ = 0.865) and specific apparent modulus (p = 0.008, $R^2$ = 0.857). For the mammalian species, no correlation with body mass is observed for apparent (p < 0.268) and specific apparent modulus (p = 0.164). The apparent and specific apparent moduli were dependent on the trabecular bone architectural indices. For the dinosaurian species, apparent elastic modulus was found to follow the equation $E_{App}$ = 0.0722 x Conn.D (p = 0.062, $R^2$ = 0.5337) and specific apparent modulus was found to follow the equation $E_{App\,Spec}$ = 0.0974 x Conn.D (p = 0.056, $R^2$ = 0.5504). For the mammalian species, apparent elastic modulus was found to follow the equation $E_{App}$ = 10.67 x Tb.Th (p < 0.001, $R^2$ = 0.9644) and specific apparent modulus was found to follow the equation $E_{App\,Spec}$ = 5.32 x Tb.Th + 4.65 x Tb.Sp (for the constants, p = 0.056 and 0.017, respectively, for the regression, p = 0.001 and $R^2$ = 0.9741)

Average tensile and average compressive principal strains are shown in Fig 5, where all strain magnitudes were all less than or equal 2,856 microstrain. For the dinosaurian models no correlation with body mass was observed for the average tensile (p = 0.403) or average compressive (p = 0.156) principal strains. Similarly, there was no correlation between body mass and the largest tensile (5,394 ± 1,750 microstrain, p = 0.668) or largest compressive (10,587 ±

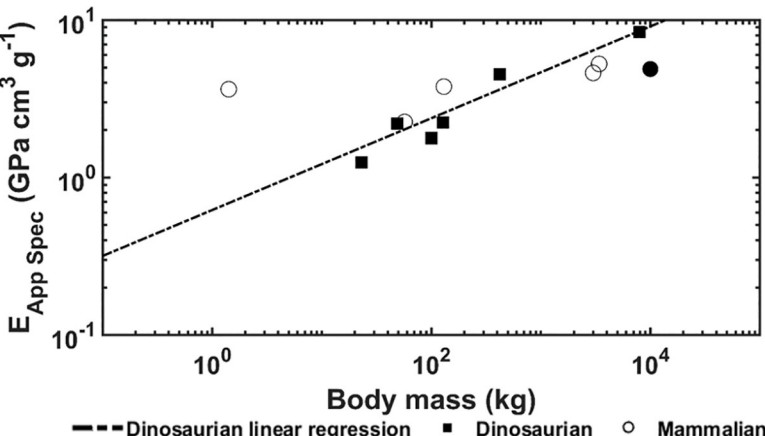

**Fig 4. Specific apparent elastic modulus versus body mass.** The solid circle indicates the mammoth.

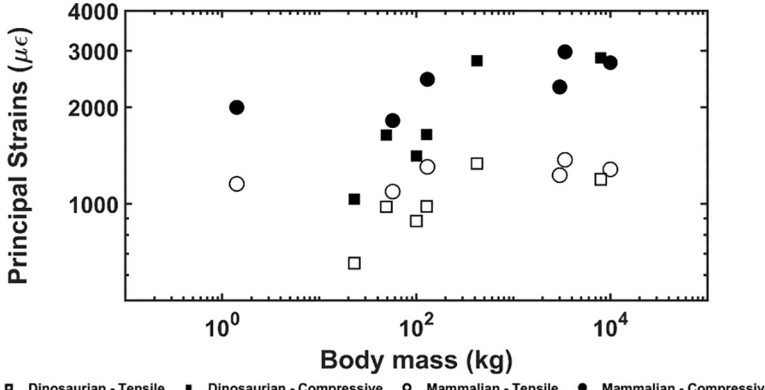

**Fig 5. Average compressive and tensile principal strains versus body mass.** Strains are shown in microstrain (με). There is no correlation between body mass and the compressive/tensile principal strains in both mammalian and dinosaurian trabecular bone.

3,099 microstrain, p = 0.122) principal strains. For the mammalian models, no correlation with body mass was found for the average tensile (p = 0.398) or average compressive principal strain (p = 0.167). Similarly, for the mammalian models, no correlation was observed between body mass and the largest tensile (4,992 ± 1,080 microstrain, p = 0.649) and compressive (10,018 ± 2,062 microstrain, p = 0.316) principal strains.

## Discussion

Allometry and mechanical performance of trabecular bone architecture of extant and extinct species (i.e., dinosaurs and mammoth) were investigated to provide framework for understanding how trabecular bone helped support extremely massive animals. Previous studies of extant mammalian and avian species found no correlation between trabecular bone volume fraction and body mass in animals ranging in body mass from mouse to elephant [8, 9]. This result is surprising since animals with greater mass require stiffer bone structures to support larger gravitational loads and apparent elastic modulus is positively correlated with bone volume fraction [10]. It is possible that the trabecular architecture of extremely massive animals was adapted to accommodate large gravitational loads while minimizing bone mass by maintaining a constant bone volume fraction. The trabecular architecture of dinosaurs has been related to locomotor behavior [19–21], but relationships between trabecular bone architectural indices and mechanical performance indices were not established. Our results show that dinosaurian trabecular bone volume fraction is positively correlated with body mass unlike what has been observed in extant mammalian and avian species previously. However, when data from mammalian and avian species is limited to trabecular bone from the femoral and tibial condyles for direct comparison to samples in this study, they too demonstrate positive correlation between bone volume fraction and animal mass. Additionally, trabecular spacing is negatively correlated with body mass while connectivity density is positively correlated with body mass in dinosaurs. These trends exhibit opposite behavior of the trends observed for extant mammalian and avian species. Despite these differences, it was found that both mammalian and dinosaurian trabecular bone architectures limit average trabecular tissue strains to under 3,000 microstrain for estimated high levels of physiological loading. Interestingly, mammalian trabecular bone was found to limit strains by increasing trabecular thickness while dinosaurian trabecular bone limits strains by increasing connectivity density.

One limitation of this study is that human trabecular bone mechanical properties were used in the finite element models because it was impossible to know the mechanical properties of the fossilized bone samples. Despite this assumption, our findings are insightful because using the same mechanical properties across all finite element models allows for direct comparison between the trabecular architectures of these animals. However, it should be recognized the fossilized samples could have had different material properties in life due to factors such as differences in mineral content. Another limitation with this study is the relatively low number of samples. This was due to the limited amount of dinosaur and mammoth bone samples available for assessing trabecular bone architecture. With that said, our results are insightful as this is the first study to assess relationships between trabecular bone architectural indices and mechanical behavior in dinosaurian species. A third limitation is that the exact mass of each species was unknown. While current estimates of species masses likely provide reasonably accurate values for the context of this study, a lack of individual sample masses limits the power of the regression analyses. Despite these limitations, we found the trabecular bone allometry in dinosaurian species exhibits allometric scaling with opposite behavior, except bone volume fraction, compared to extant mammalian and avian species, apparent trabecular bone stiffness is positively correlated with body mass in dinosaurian species, and dinosaurian and mammalian trabecular bone architecture limits average strains to below 3,000 microstrain. These findings provide insight into how trabecular bone in the distal femur and proximal tibia adapted to support extremely large body masses. A fourth limitation is the mostly unknown age of the of the fossil specimen of this study. Based on absolute size, sampled *Edmontosaurus regalis* specimens fall into the adult ontogenetic stage. True growth curves based upon histological sampling have not been assembled for this dinosaur taxon and are well beyond the scope of this work. Size is an accurate indicator of ontogenetic stage in hadrosaur dinosaurs [66], and these samples were restricted to elements in the upper half of size, often associated with somatic maturity in other dinosaur taxa. Similarly, using the basis of the specimens' large size, the theropod dinosaurs were presumed to all derive from adult animals [20], but assessing ontogenetic status in any dinosaur is a notoriously tricky issue [31]. Based on prior histological work by one of the authors, it was suggested that the *Edmontosaurus annectens* used in this study were late stage juveniles [67, 68]. We recognize that the individuals represented by the sample were not fully grown but given that these individuals do represent late stage juveniles, as opposed to very young individuals. An additional limitation is that dinosaur sex is not currently attainable. Sex determination in dinosaurs is difficult in the absence of medullary bone [21]. Dinosaur taxa for which large samples sizes are known (including Edmontosaurus in this study) do not demonstrate any statistically discernable population dimorphism [69]. Furthermore, there has yet to be demonstrated a convincing argument for sexual dimorphism in any extinct, non-avian dinosaur [70].

The allometric scaling relationships show how the trabecular bone architectural indices scale with body mass in dinosaurian, mammalian, and avian species. Unlike previous studies [8, 39], the present research focused only on the trabecular bone from the distal femur and proximal tibia which uncovered some interesting differences. First, the trabecular bone volume fraction in these locations shows positive correlation with body mass for dinosaurian, mammalian, and avian species (Fig 6). These results contrast previous findings that showed no correlation between bone volume fraction and body mass when looking at numerous skeletal locations together [8, 9]. Skeletal locations in previous studies included the calcaneus, femoral condyles, head, trochanter, and neck, proximal and distal tibia, vertebrae, radius, ulna, iliac crest, and humerus. It is possible that our results for the distal femur and proximal tibia differ from previous results due to differences in mechanical loading at each location. Trabecular bone in the distal femur and proximal tibia have been shown to have similar architectural

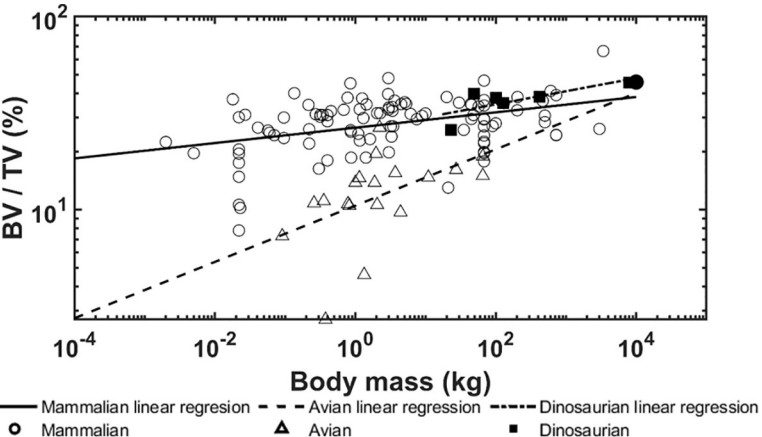

**Fig 6. Logarithmically scaled plots of the bone volume fraction (BV/TV) versus body mass.** Pairwise comparisons indicate the dinosaur regression slope is not different from the mammalian ($p = 0.352$) and avian ($p = 0.695$) slopes. The solid circle indicates the mammoth.

properties [37] and therefore may have adapted differently than trabecular architectures in other bones to accommodate their specific mechanical loading conditions. Furthermore, it is known that trabecular bone architectural indices scale [9] with body mass at higher rates than cortical bone increases thickness [71], thereby indicating that trabecular bone would play a larger role in load sharing at larger body masses. It is currently known that trabecular bone in the femoral neck experiences 76–89% of the incident load [72]. Second, no correlation between trabecular thickness and body mass was observed for dinosaurs while a positive correlation was observed for mammalian and avian species (Fig 7). Previously, it has been shown that larger body mass animals have greater trabecular thickness to prevent individual trabeculae from being overly strained [9]. The fact that dinosaur trabeculae do not follow this trend is an interesting result and suggests other trabecular bone indices may adapt to provide increased mechanical competence instead. In support of this theory, we have shown that trabecular spacing is negatively correlated with body mass, while trabecular number and connectivity density

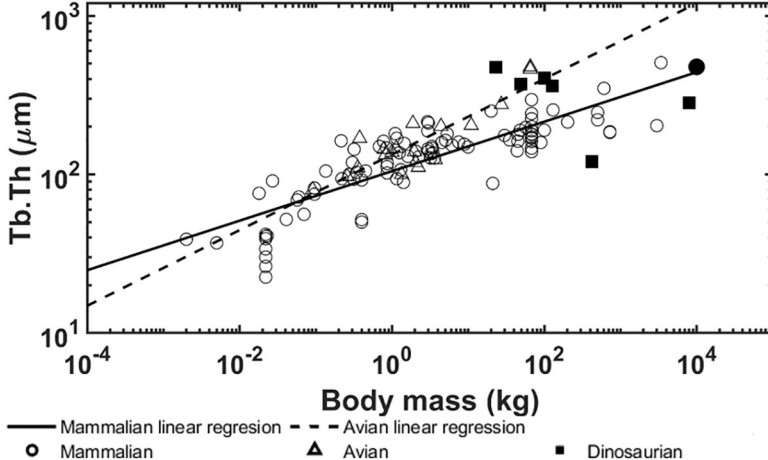

**Fig 7. Logarithmically scaled plots of the trabecular thickness (Tb.Th) versus body mass.** Pairwise comparisons indicate the dinosaur regression slope is different from the mammalian ($p < 0.001$) and avian ($p < 0.001$) slopes. The solid circle indicates the mammoth.

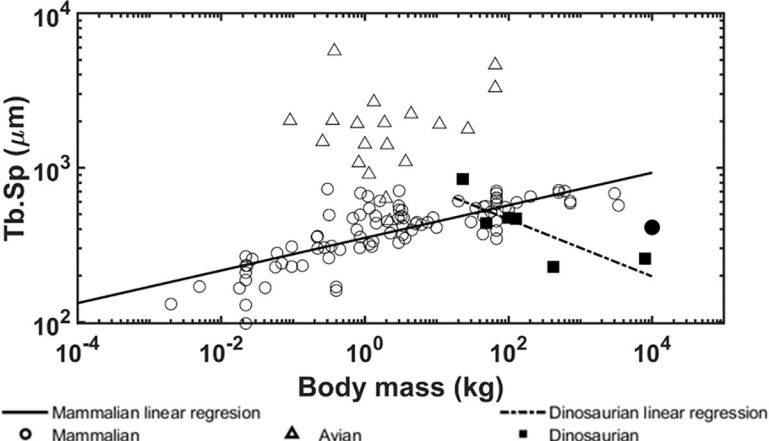

**Fig 8. Logarithmically scaled plots of the trabecular spacing (Tb.Sp) versus body mass.** Pairwise comparisons indicate the dinosaur regression slope is different from the mammalian ($p < 0.001$) and avian ($p = 0.007$) slopes. The solid circle indicates the mammoth.

are positively correlated with body mass in the dinosaurian species (Figs 8, 9 and 10). These trends are opposite of those observed for the avian and mammalian species. Thus, it appears that, as dinosaurs grow larger, decreased trabecular spacing and increased connectivity density and trabecular number provide sufficient mechanical stability while maintaining a relatively constant trabecular thickness. These trends are further elucidated with results from the finite element models.

Computational models demonstrated positive correlations between body mass and trabecular bone apparent and specific apparent moduli for the dinosaurian species as expected (Figs 3 and 4). These findings confirm the hypothesis that stiffer trabecular architectures are developed as animal size increases to support greater mechanical loads. Interestingly, this contrasts previous findings which showed no correlation between animal size and apparent modulus of trabecular bone in mammalian species [9]. For dinosaurian species, the apparent and specific apparent moduli are both dependent only on connectivity density. For the mammalian species, trabecular bone apparent modulus is dependent only on trabecular thickness, but specific

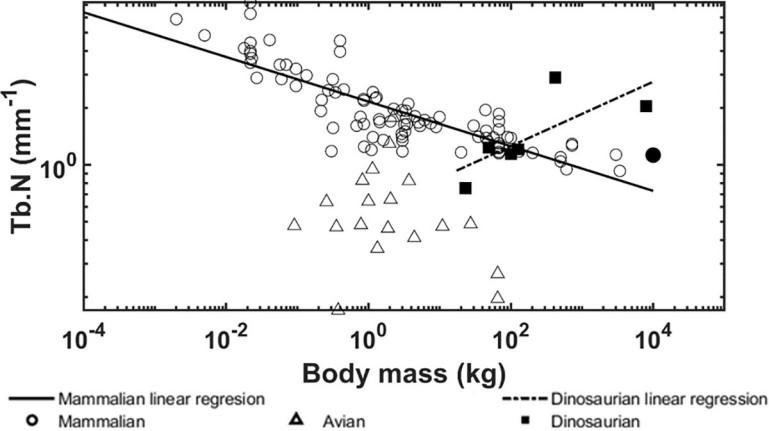

**Fig 9. Logarithmically scaled plots of the trabecular number (Tb.N) versus body mass.** Pairwise comparisons indicate the dinosaur regression slope is different from the mammalian ($p < 0.001$) and avian ($p = 0.004$) slopes. The solid circle indicates the mammoth.

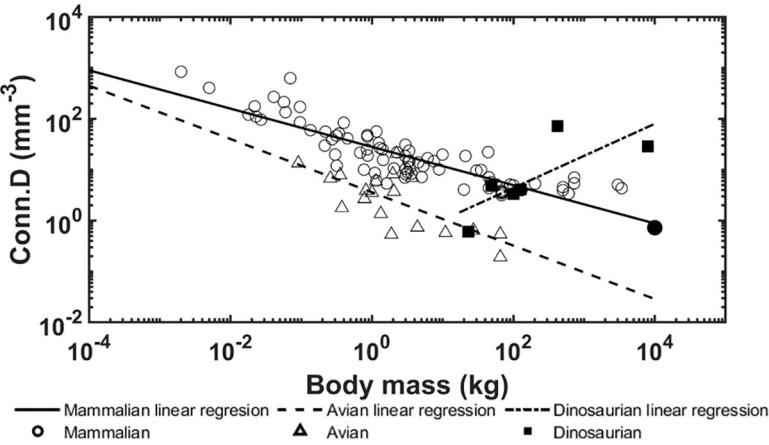

**Fig 10. Logarithmically scaled plots of connectivity density (Conn.D) versus body mass.** Pairwise comparisons indicate the dinosaur regression slope is different from the mammalian (p < 0.001) and avian (p < 0.001) slopes. The solid circle indicates the mammoth.

apparent modulus is dependent on trabecular thickness and spacing together. The dependence of trabecular bone stiffness on trabecular thickness and connectivity density is not novel [10, 16–18]. However, it is interesting that increases in bone stiffness were achieved through increased connectivity density in dinosaurs but increased trabecular thickness in mammals. The reason for this is currently unclear, but one explanation could be that high connectivity is a more efficient stiffening mechanism than increased trabecular thickness, especially for the exceptional loads produced by the mass of the largest animals. This idea is analogous to the load sharing utilized by trusses to achieve weight reduction in structural design and may have been used by dinosaurs to constrain whole bone weight and trabecular bone tissue strains.

Despite the allometric scaling of the apparent and specific apparent moduli, we found that the average principal strain magnitudes were not correlated with body mass. Furthermore, average principal strain magnitudes were limited to 3,000 microstrain for all samples in this study. Similar limits have been previously observed for mammalian bone from a variety of species during routine activities such as running, jumping, walking, and chewing [4, 73–77]. Strain limits are achieved as bone remodels in response to mechanical loading [24, 78, 79]. The remodeling process limits high strains to decrease the risk of fracture [76, 80] and low strains to avoid excess bony material in areas where it is mechanically unnecessary [81]. Previous studies on the trabecular architecture in mammalian species suggested that trabecular thickness increased with increasing body mass in order to modulate the strains experienced in individual trabeculae [9]. In the case of dinosaurian species, it appears that an equivalent result is achieved by increasing connectivity density instead of trabecular thickness. This result is similar to what was observed for the apparent and specific apparent moduli of each species. It is unclear why dinosaur bone adapted to have higher connectivity density instead of increased trabecular thickness; however, as mentioned previously, it's possible that this mechanism of strain modulation more efficiently balances the structures mechanical competence and weight. Unfortunately, testing this idea is beyond the scope of this paper. It may be possible for future studies to carefully design a porous structure where trabecular thickness and connectivity density can be carefully controlled such that their individual effects can be ascertained.

The present study provides evidence of how trabecular architecture supported large body masses. However, it must be considered that dinosaurian trabecular tissue may differ from extant mammalian trabecular tissue on a compositional level which would have implications

for the mechanical behavior of this tissue [82–93]. However, due to the fossilization of dinosaur bones, this cannot be accurately assessed. Either way, using the same material properties in direct comparisons of bone architectures showed that dinosaur trabecular bone apparent modulus and bone volume fraction are positively correlated with body mass. Additionally, the trabecular bone apparent modulus shows strong dependence on trabecular bone connectivity density in dinosaurian species. Taken together, it is concluded that the trabecular architecture in dinosaurs evolved to maintain bone stiffness and modulate strain levels to prevent failure across a wide range of body masses. Our data also demonstrate that changes in connectivity density were the primary mechanism for dinosaur bone adaptation. However, at this point, it is unclear why dinosaurs altered connectivity density to achieve this result instead of adjusting trabecular thickness like mammals. We suggest that increasing connectivity is a more efficient stiffening mechanism than increasing strut thickness for animals of this extraordinary size. This would have allowed for sufficient mechanical competence to be achieved with less bone material (i.e. minimizing the metabolic cost of maintaining and transporting bony material). These findings have potential implications for novel bioinspired designs of stiff and lightweight structures that could be used in aerospace, construction, or vehicular applications.

## Acknowledgments

The authors acknowledge Dr. Peter Bishop at the Royal Veterinary College (London, England, UK) for his assistance in procuring CT scans of *Troodontidae*, *Caenagnathidae*, *Ornithomimidae*, and *Falcarius utahensis*.

The authors thank Dr. Don Henderson and Mr. Brandon Strilisky at the Royal Tyrrell Museum of Palaeontology (Drumheller, Alberta, CA) for allowing access to their *Caenagnathidae* (TMP 1986.036.0323) and *Ornithomimidae* (TMP 1999.055.0337) specimen CT scans.

The authors also thank Dr. John Scannella and Ms. Amy Atwater at the Museum of the Rockies at Montana State University and the land management agencies from whose land the specimens were collected, for allowing us access of their *Troodontidae* (MOR 748) specimen CT scans.

Courtesy of Natural History Museum of Utah, UMNH VP 12360, the authors thank Dr. Randall B. Irmis at the Natural History Museum of Utah for allowing us access to their *Falcarius utahensis* specimen CT scans.

The authors acknowledge Dr. Marieka Arksey in the Department of Anthropology at the University of Wyoming and the University of Wyoming Archaeological Repository for allowing us to collect the *Mammuthus columbi* specimen sample (48WA322-9). Additionally, the authors thank Ms. Rebecca Brower at the Washakie Museum in Worland Wyoming for coordinating with the authors to collect a sample from the mammoth fossil.

Specimens sampled at the Denver Museum of Nature & Science are part of the Hankla Family Collection, generously donated for the preservation and promotion of science.

The authors thank Dr. Ann Hess in the Department of Statistics at Colorado State University for her assistance with statistical methods utilized in this study.

The authors thank Alison Doherty and Jason Hinrichs in the Mechanical Engineering Department at Colorado State University for assistance developing the uCT scanning protocols for fossilized bone and for developing the methods for coring the *Edmontosaurus* samples, and Steve Johnson in the Mechanical Engineering Department at Colorado State University for design assistance, manufacturing of the mobile core drilling device used in this study and helping collect the *Mammuthus columbi* trabecular core.

*Edmontosaurus annectens* specimens were collected under BLM permit number AA−86367, with administrative support from the Arctic Management Unit of the Bureau of Land Management.

All necessary permits were obtained for the described study, which complied with all relevant regulations.

## Author Contributions

**Conceptualization:** Trevor G. Aguirre, Seth W. Donahue.

**Data curation:** Trevor G. Aguirre, Tim W. Seek, Anthony R. Fiorillo, Joseph J. W. Sertich.

**Formal analysis:** Trevor G. Aguirre.

**Investigation:** Trevor G. Aguirre.

**Methodology:** Trevor G. Aguirre, Aniket Ingrole, Tim W. Seek.

**Project administration:** Seth W. Donahue.

**Software:** Trevor G. Aguirre.

**Supervision:** Seth W. Donahue.

**Validation:** Trevor G. Aguirre.

**Visualization:** Trevor G. Aguirre.

**Writing – original draft:** Trevor G. Aguirre.

**Writing – review & editing:** Trevor G. Aguirre, Luca Fuller, Anthony R. Fiorillo, Joseph J. W. Sertich, Seth W. Donahue.

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
