## [Decision Letter · Decision Letter 0]

11 Jun 2020

PONE-D-20-11333

Differing trabecular bone architecture in dinosaurs and mammals contribute to stiffness and limits on bone strain

PLOS ONE

Dear Dr. Aguirre,

Thank you for submitting your manuscript to PLOS ONE. After careful consideration, we feel that it has merit but does needs clarifications in certain areas.  Also, the limitations need to be included in the discussion.  I am hoping to accept the manuscript subjected to these minor revisions.

We look forward to receiving your revised manuscript.

Kind regards,

Deepak Vashishth, Ph.D.

Academic Editor

PLOS ONE

Journal Requirements:

2. Please amend your Financial disclosure statement in the online submission form to declare sources of funding.

3. Please clarify in your Data availability statement how other researchers can obtain the data used in, and specifically obtained for, this study.

4. In your manuscript, please provide additional information regarding the specimens used in your study. Ensure that you have reported specimen numbers and complete repository information, including museum name and geographic location.

For more information on PLOS ONE's requirements for paleontology and archaeology research, see https://journals.plos.org/plosone/s/submission-guidelines#loc-paleontology-and-archaeology-research.

5. Please ensure that you refer to Figure 6 in your text as, if accepted, production will need this reference to link the reader to the figure.

Additional Editor Comments (if provided):

Reviewers' comments:

Reviewer's Responses to Questions

**Comments to the Author**

1. Is the manuscript technically sound, and do the data support the conclusions?

Reviewer #1: Yes

Reviewer #2: Partly

2. Has the statistical analysis been performed appropriately and rigorously? 

Reviewer #1: Yes

Reviewer #2: Yes

3. Have the authors made all data underlying the findings in their manuscript fully available?

Reviewer #1: Yes

Reviewer #2: Yes

4. Is the manuscript presented in an intelligible fashion and written in standard English?

Reviewer #1: Yes

Reviewer #2: Yes

5. Review Comments to the Author

Reviewer #1: General:

This is an interesting report, with only a few minor issues as noted below.

Specific:

Abstract Line 1, "Dinosaurs were exceptionally large animals": This isn't really true. Some of them were enormous, but many (including some in the present study where the smallest was estimated 23kg) weren't very big. Please correct this.

Line 192, "the linear region": This is misleading since the entire analysis was done using linear elastic finite element analysis. The resulting "load-displacement" curves are straight for all strain. The apparent stiffness was likely calculated as the ratio of resultant force divided by applied displacement. Please correct this.

Lines 239-257: What statistical software was used for the linear regressions, pairwise comparisons, and stepwise regressions?

Line 241: What statistical test was used for the pairwise comparisons?

Line 243/253: What inclusion criteria were used to accept for reject independent variables in the stepwise regressions? Also, were these linear stepwise regressions?

Line 257: It took me a very long time for find the sample size for dinosaur specimens (Lines 123-127) and I could not find sample sizes for the Mammals or Avian species. I'd suggest adding sample sizes to Table 1 or somewhere in the body of the paper. Were averages or raw data points used for the other species?

Reviewer #2: This study investigates how trabecular bone architectural indices changed with body mass in bone samples from the distal femur and proximal tibia of dinosaurs ranging in body mass from 23-8,000 kg. Using CT, uCT, and finite element modeling, trabecular bone mechanical behavior was investigated. Dinosaurian trabecular bone volume fraction was positively correlated with body mass similar to what was observed for extant mammalian species, while trabecular spacing, number, and connectivity density in dinosaurs was negatively correlated with body mass, exhibiting opposite behavior from extant mammals. Trabecular bone apparent modulus was positively correlated with body mass in dinosaurian species, while no correlation was observed for mammalian species. Additionally, trabecular bone tensile and compressive principal strains were not correlated with body mass in mammalian or dinosaurian species. This is the first study to assess relationships between trabecular bone architectural indices and mechanical behavior in dinosaurian species. However, there are a number of limitations with study design and methodology.

A strength of the paper is that it focuses on the femur/tibia and not other anatomical locations which may different mechanical loading environments. This important to the context of the results. Yet, limitations reduce the impact.

1. P.5, Section 2.1, Species analyzed: Only 2 species of dinosaur, both hadrosaurs, were analyzed. While the size difference was large, a greater variety of species with differing physical activity (based on location, diet, etc.), diet, and size would improve the statistical significance of the results.

a. What were the age estimates of the dinosaurs (i.e., were these considered to be adult, adolescent, etc.)?

b. What is the assumed physical activity of these dinosaurs (hunters, climbers, swimmers, flyers, etc.)?

c. What was the sex of the animals tested? If sex was known, sexual dimorphism could influence results.

2. P. 9, Section 2.4, Material properties: It is understood that material properties cannot be determined from fossilized specimens. However, could the analysis be repeated with different material properties to see if bone mineral density could influence the results?

3. P. 22, ll. 358-361. “Trabecular bone in the distal femur and proximal tibia have been shown to have similar architectural properties and therefore may have adapted differently than trabecular architectures in other bones to accommodate their specific mechanical loading conditions.” Is it known how trabecular bone architecture correlates with cortical bone thickness? Thicker cortical bone could affect the amount of loading that the trabecular bone sees despite similar physical activity loading.

4. P. 24, ll. 386-388. “…high connectivity is a more efficient stiffening mechanism than increased trabecular thickness, especially for the exceptional loads produced by the mass of the largest animals.” This is testable. Trabecular thickness can be artificially increased in the FE models for the lower body mass species and can be compared to the mechanical properties obtained with increased trabecular connectivity. It’s possible that it is less efficient as an slight increase in thickness can result in an increase in stiffness with less bone mass added than if connectivity density is increased as is.

6. PLOS authors have the option to publish the peer review history of their article (what does this mean?). If published, this will include your full peer review and any attached files.

Reviewer #1: No

Reviewer #2: No

---

## [Author Response · Author response to Decision Letter 0]

10 Jul 2020

Responses to Reviewers

Journal: PLOS One

Manuscript Number: PONE-D-20-11333

Manuscript Title: Differing trabecular bone architecture in dinosaurs and mammals contribute to stiffness and limits on bone strain

Authors: Trevor G. Aguirre, Aniket Ingrole, Luca Fuller, Tim W. Seek, Anthony R. Fiorillo, Joseph J.W. Sertich, Seth W. Donahue

We would like to thank the reviewers for taking the time to evaluate our manuscript and for providing us with valuable feedback. Our responses to your comments and questions are provided below in blue. The corresponding revisions are marked-up with red font below and in the manuscript. 

Responses to Reviewer’s Comments

Editor: 

1 - Please ensure that your manuscript meets PLOS ONE's style requirements, including those for file naming.

Response: Thank you for your feedback on formatting. The text has been modified to reflect the required formatting described at the links you have kindly provided. Since the entirety of the manuscript required reformatting the revised text is not provided here. 

2- Please amend your Financial disclosure statement in the online submission form to declare sources of funding.

Response: Thank you for your comment. The financial disclosure statement has been amended in the online submission form. 

3- Please clarify in your Data availability statement how other researchers can obtain the data used in, and specifically obtained for, this study.

Response: Thank you for your comment. Our data availability statement has been revised to help researchers obtain the data used in this study. 

Revised Data availability statement:

All relevant data are within the manuscript and its Supporting Information files.

CT scans of extant species of this study are available online at:https://figshare.com/articles/X-ray_microtomography_images_of_trabecular_bone_from_the_femoral_head_and_condyle_of_18_avian_72_mammalian_and_one_crocodilian_species_/7257179

Architectural index data for extant species are available in the Supplemental Information for references 8 and 9.

CT scans of the extant species can be obtained by contacting the relevant researchers in the acknowledgements section.

4- In your manuscript, please provide additional information regarding the specimens used in your study. Ensure that you have reported specimen numbers and complete repository information, including museum name and geographic location.

Response: Thank you for pointing this out. The text before Table 1 has been revised to reflect the number of specimens used in this study. The legend below Table 1 has been updated to reflect the city, state/province, and country for each museum referenced in the study. 

Revised text (Line 117):

The species used in the finite element models of this study were chosen to cover a wide range of body masses, from 1 to ~10,000 kg, and are listed in Table 1. CT and µCT scans from previous studies were used for some species as indicated in Table 1. For Mammuthus columbi and Edmontosaurus, new bone samples were obtained and scanned with µCT for the current study. One sample was used for each specimen listed in Table 1.

Revised Table 1 legend (Line 124):

University Museum of Zoology Cambridge (UMZC), Cambridge, United Kingdom, Europe., Royal Veterinary College (RVC), London, United Kingdom, Europe. 

Museum of the Rockies (MOR), Bozeman, Montana, United States of America.

Royal Tyrrell Museum of Palaeontology (TMP), Drumheller, Alberta, Canada. 

Natural History Museum of Utah (UMNH), Salt Lake City, Utah, United States of America.

Perot Museum of Nature and Science (1DMNH), Dallas, Texas, United States of America.

Denver Museum of Nature & Science (2DMNH), Denver, Colorado, United States of America.

University of Wyoming Archaeological Repository (UWAR), Laramie, Wyoming, United States of America. 48WA is the archaeological site identification code per the Smithsonian trinomial system. 

 

5- Please ensure that you refer to Figure 6 in your text as, if accepted, production will need this reference to link the reader to the figure.

Response: Thank you for pointing this out and we apologize for this oversight. The text has been modified to link the reader to this figure. 

Revised text (Lines 404-410):

In support of this theory, we have shown that trabecular spacing is negatively correlated with body mass, while trabecular number and connectivity density are positively correlated with body mass in the dinosaurian species (Figure 5, Figure 6, and Figure 7). These trends are opposite of those observed for the avian and mammalian species. Thus, it appears that, as dinosaurs grow larger, decreased trabecular spacing and increased connectivity density and trabecular number provide sufficient mechanical stability while maintaining a relatively constant trabecular thickness. 

 

Reviewer #1: 

1- Abstract Line 1, "Dinosaurs were exceptionally large animals": This isn't really true. Some of them were enormous, but many (including some in the present study where the smallest was estimated 23kg) weren't very big. Please correct this.

Response: Thank you for the comment. The main text has been revised to more clearly represent that only the largest dinosaurs were enormous

Revised text (Line 32):

The largest dinosaurs were enormous animals whose body mass placed massive gravitational loads on their skeleton. Previous studies investigated dinosaurian bone strength and biomechanics, but the relationships between dinosaurian trabecular bone architecture and mechanical behavior has not been studied.

2- Line 192, "the linear region": This is misleading since the entire analysis was done using linear elastic finite element analysis. The resulting "load-displacement" curves are straight for all strain. The apparent stiffness was likely calculated as the ratio of resultant force divided by applied displacement. Please correct this.

Response: Thank you for the comment. The main text has been revised to more clearly describe the calculation.

Revised text (Lines 208-211):

The effect of the trabecular bone architecture on the apparent elastic modulus (E App) was determined by dividing the peak apparent stress by the applied displacement for each finite element model. The peak apparent stress was computed by dividing the peak load by the specimen area, which was computed from the bulk dimensions of each finite element model.

3- Lines 239-257: What statistical software was used for the linear regressions, pairwise comparisons, and stepwise regressions?

Response: Thank you for your question. The software we used was Minitab. We have included the version number of the software to make this clearer to the reader. 

Revised text (Line 272):

Linear regressions, pairwise comparisons, and stepwise regressions were computed using Minitab (version 18). 

4- Line 241: What statistical test was used for the pairwise comparisons?

Response: Thank you for question. The statistical test was a Tukey post-hoc test. The text has been revised.

Revised text (Line 259):

Pairwise comparisons were made between regression slopes of the mammalian, avian, and dinosaurian species using a Tukey post-hoc Test.

5- Line 243/253: What inclusion criteria were used to accept for reject independent variables in the stepwise regressions? Also, were these linear stepwise regressions

Response: Thank you for question. Independent variables were rejected from the stepwise regression if p > 0.1. This main text has been revised to include this. Furthermore, the text has been revised to tell the reader that the stepwise regressions were linear. 

Revised text (Lines 260-266):

Linear stepwise regressions were used to determine if the trabecular bone architectural indices predict the apparent and specific apparent elastic moduli. The candidate independent variables were Tb.Th, Tb.Sp, and Conn.D, and the dependent variables were apparent elastic modulus and specific apparent elastic modulus. Trabecular number and bone volume fraction were excluded from stepwise regression models to avoid collinearity since both of these parameters are dependent on trabecular thickness and trabecular spacing (35). Independent variables were rejected if p > 0.1. 

6- Line 257: It took me a very long time for find the sample size for dinosaur specimens (Lines 123-127) and I could not find sample sizes for the Mammals or Avian species. I'd suggest adding sample sizes to Table 1 or somewhere in the body of the paper. Were averages or raw data points used for the other species?

Response: We have modified the manuscript to include a specific statement that only one sample was used per species, except the two groups of hadrosaur. 

Revised text (Line 117):

The species used in the finite element models of this study were chosen to cover a wide range of body masses, from 1 to ~10,000 kg, and are listed in Table 1. CT and µCT scans from previous studies were used for some species as indicated in Table 1. For Mammuthus columbi and Edmontosaurus, new bone samples were obtained and scanned with µCT for the current study. One sample was used for each specimen listed in Table 1.

Since there were more than one sample in the two hadrosaur groups, average values from each group were used in the regressions. This was indicated in the original manuscript submission in the “Statistical Analyses” section and is provided below for reference. 

Line2 272-274 in revised manuscript

Due to the imbalance between the numbers of dinosaurian samples the average values for Edmontosaurus regalis and Edmontosaurus sp. were used in all regressions.

 

Reviewer #2: 

1- P.5, Section 2.1, Species analyzed: Only 2 species of dinosaur, both hadrosaurs, were analyzed. While the size difference was large, a greater variety of species with differing physical activity (based on location, diet, etc.), diet, and size would improve the statistical significance of the results.

a. What were the age estimates of the dinosaurs (i.e., were these considered to be adult, adolescent, etc.)?

b. What is the assumed physical activity of these dinosaurs (hunters, climbers, swimmers, flyers, etc.)?

c. What was the sex of the animals tested? If sex was known, sexual dimorphism could influence results.

Response: Thank you for your questions and comments. We studied two species of hadrosaur and four species of theropod (Troodontidae, Caenagathidae, Ornithomimidae, Falcarius Utahensis) as indicated in Table 1.

a. We are unable to determine the age of the dinosaurs. Based on absolute size, the sampled Edmontosaurus regalis specimens fall into the adult ontogenetic stage. True growth curves based upon histological sampling have not been assembled for this dinosaur taxon and are well beyond the scope of this contribution. Size is an accurate indicator of ontogenetic stage in hadrosaur dinosaurs, and this sample was restricted to elements in the upper half of size, often associated with somatic maturity in other dinosaur taxa. Similarly, using the basis of the specimens' large size, the theropod dinosaurs were presumed to all derive from adult animals, but assessing ontogenetic status in any dinosaur is a notoriously tricky issue. Based on prior histological work by one of the authors, it was suggested that the Edmontosaurus sp. used in this study were late stage juveniles. We recognize that the individuals represented by the sample were not fully grown but given that these individuals do represent late stage juveniles, as opposed to very young individuals.

b. Hadrosaurid dinosaurs (aka, duckbills) are largely quadrupedal, terrestrial walkers, though they are able to move bipedally when necessary, presumably to access high-growing vegetation or to escape predation. This is based upon a combination of trackway evidence, morphology, and locomotion modeling studies. All theropod species involved were bipedal walkers and runners.

c. Dinosaur sex is not attainable in these samples. Sex determination in dinosaurs is difficult in the absence of medullary bone. Dinosaur taxa for which large samples sizes are known (including Edmontosaurus in this study) do not demonstrate any statistically discernable population dimorphism. Furthermore, there has yet to be demonstrated a convincing argument for sexual dimorphism in any extinct, non-avian dinosaur. 

The manuscript has been revised in two areas to address the above questions. 

Revised text (Lines 117-123):

The species used in the finite element models of this study were chosen to cover a wide range of body masses, from 1 to ~10,000 kg, and are listed in Table 1. CT and µCT scans from previous studies were used for some species as indicated in Table 1. For Mammuthus columbi and Edmontosaurus, new bone samples were obtained and scanned with µCT for the current study. The mammalian species utilized in this study are all quadrupedal whose primary form of locomotion is walking and running. The hadrosaur specimens were largely quadrupedal but capable of bipedal motion when necessary [26–28]. All theropods were bipedal. Furthermore, the hadrosaur and theropods primary form of locomotion is walking and running [29, 30]. Sex is unknown for all species, the mammals were skeletally mature [9], and age or skeletal maturity is unknown for all dinosaurian species as this is difficult to determine from the fossil record [31].

Revised text (Lines 363-380) :

A fourth limitation is the unknown ages of the fossil specimens of this study. Based on absolute size, sampled Edmontosaurus regalis specimens fall into the adult ontogenetic stage. True growth curves based upon histological sampling have not been assembled for this dinosaur taxon and are well beyond the scope of this work. Size is an accurate indicator of ontogenetic stage in hadrosaur dinosaurs [66], and these samples were restricted to elements in the upper half of size, often associated with somatic maturity in other dinosaur taxa. Similarly, using the basis of the specimens' large size, the theropod dinosaurs were presumed to all derive from adult animals [20], but assessing ontogenetic status in any dinosaur is a notoriously tricky issue [31]. Based on prior histological work by one of the authors, it was suggested that the Edmontosaurus annectens used in this study were late stage juveniles [67, 68]. We recognize that the individuals represented by the sample were not fully grown but given that these individuals do represent late stage juveniles, as opposed to very young individuals. An additional limitation is that dinosaur sex is not currently attainable. Sex determination in dinosaurs is difficult in the absence of medullary bone [21]. Dinosaur taxa for which large samples sizes are known (including Edmontosaurus in this study) do not demonstrate any statistically discernable population dimorphism [69]. Furthermore, there has yet to be demonstrated a convincing argument for sexual dimorphism in any extinct, non-avian dinosaur [70]. 

2- P. 9, Section 2.4, Material properties: It is understood that material properties cannot be determined from fossilized specimens. However, could the analysis be repeated with different material properties to see if bone mineral density could influence the results?

Response: Thank you for your question. The analysis could be repeated but the trends would remain the same. If a lower elastic modulus were assigned to the finite element models, we would observe a proportionally lower measured apparent elastic modulus for all specimen. The trend would remain the same for the principal strains because when the eigenvalues are computed, the elastic modulus is divided out, assuming a linear isotropic material model. The largest difference would come from knowing the materials properties for all specimen (avian, extant/extinct mammalian, and dinosaurian). The assumption that material properties are the same for all the finite element models allows us to directly compare the trabecular architectures, which was the main goal of the study. This limitation of our study is addressed in the discussion in lines 345-351 and lines 451-454. 

 Lines 345-351 in revised manuscript:

One limitation of this study is that human trabecular bone mechanical properties were used in the finite element models because it was impossible to know the mechanical properties of the fossilized bone samples. Despite this assumption, our findings are insightful because using the same mechanical properties across all finite element models allows for direct comparison between the trabecular architectures of these animals. However, it should be recognized the fossilized samples could have had different material properties in life due to factors such as differences in mineral content.

Lines 451-454 in revised manuscript:

However, it must be considered that dinosaurian trabecular tissue may differ from extant mammalian trabecular tissue on a compositional level which would have implications for the mechanical behavior of this tissue [75–86].

3- P. 22, ll. 358-361. “Trabecular bone in the distal femur and proximal tibia have been shown to have similar architectural properties and therefore may have adapted differently than trabecular architectures in other bones to accommodate their specific mechanical loading conditions.” Is it known how trabecular bone architecture correlates with cortical bone thickness? Thicker cortical bone could affect the amount of loading that the trabecular bone sees despite similar physical activity loading.

Response: Thank you for your question. To the best of our knowledge there has not been investigation into potential correlation between trabecular architecture and cortical bone thickness with body mass in a single study. Studies have shown that trabecular thickness and spacing are positively correlated with body mass [1] with scaling exponents of 0.429 and 0.407, respectively. Furthermore, it is known that that cortical thickness is positively correlated with body mass [2] and has scaling exponents between 0.235 – 0.298. Taken, together this indicates that trabeculae get thicker and further apart at a faster rate than cortical thickness does as body mass increases, i.e. cortical thickness is negatively correlated with trabecular thickness and spacing. Thinning of cortical bone would cause increased loading on trabecular bone, where it is currently known that trabecular bone in the femoral neck experiences 76-89% of the incident load [3]. We have added this to the text. 

Revised text (Lines 395-398) :

Furthermore, it is known that trabecular bone architectural indices scale [9] with body mass at higher rates than cortical bone increases thickness [71], thereby indicating that trabecular bone would play a larger role in load sharing at larger body masses. It is currently known that trabecular bone in the femoral neck experiences 76-89% of the incident load [72].

4- P. 24, ll. 386-388. “…high connectivity is a more efficient stiffening mechanism than increased trabecular thickness, especially for the exceptional loads produced by the mass of the largest animals.” This is testable. Trabecular thickness can be artificially increased in the FE models for the lower body mass species and can be compared to the mechanical properties obtained with increased trabecular connectivity. It’s possible that it is less efficient as an slight increase in thickness can result in an increase in stiffness with less bone mass added than if connectivity density is increased as is.

Response: Thank you for your comment. Though we agree that this would be beneficial to test, we believe this hypothesis is not testable within the confines of the presented work. To investigate your suggestion, we increased the trabecular thickness of one of the FEM cubes of this study. When this was done, it was found that there was decreased trabecular spacing, increased bone volume fraction, and decreased connectivity density. Since these variables are not easily decoupled in the CT scans it is difficult to test this hypothesis without conflating the effects of the increased bone volume fraction, decreased trabecular spacing, and decreased connectivity density. Since the readership may have the same question, we have modified the discussion to include a brief explanation of why we were unable to further investigate the effect of connectivity density and trabecular thickness on trabecular bone stiffness. Furthermore, we provided a description to help inform future research directions in this area. 

Revised text (Lines 446-449):

Unfortunately, testing this idea is beyond the scope of this paper. It may be possible for future studies to carefully design a porous structure where trabecular thickness and connectivity density can be carefully controlled such that their individual effects can be ascertained.

 

Works cited

[1] M. Doube, M. M. Klosowski, A. M. Wiktorowicz-Conroy, J. R. Hutchinson, and S. J. Shefelbine, “Trabecular bone scales allometrically in mammals and birds,” Proceedings of the Royal Society B: Biological Sciences, vol. 278, no. 1721, pp. 3067–3073, Oct. 2011, doi: 10.1098/rspb.2011.0069.

[2] S. Z. M. Brianza, P. D’Amelio, N. Pugno, M. Delise, C. Bignardi, and G. Isaia, “Allometric scaling and biomechanical behavior of the bone tissue: An experimental intraspecific investigation,” Bone, vol. 40, no. 6, pp. 1635–1642, Jun. 2007, doi: 10.1016/j.bone.2007.02.013.

[3] S. Nawathe, B. P. Nguyen, N. Barzanian, H. Akhlaghpour, M. L. Bouxsein, and T. M. Keaveny, “Cortical and trabecular load sharing in the human femoral neck,” Journal of Biomechanics, vol. 48, no. 5, pp. 816–822, Mar. 2015, doi: 10.1016/j.jbiomech.2014.12.022.

---

## [Editor Report · Decision Letter 1]

20 Jul 2020

Differing trabecular bone architecture in dinosaurs and mammals contribute to stiffness and limits on bone strain

PONE-D-20-11333R1

Dear Dr. Aguirre,

We’re pleased to inform you that your manuscript has been judged scientifically suitable for publication and will be formally accepted for publication once it meets all outstanding technical requirements.

Kind regards,

Deepak Vashishth, Ph.D.

Academic Editor

PLOS ONE
---

## [Editor Report · Acceptance letter]

23 Jul 2020

PONE-D-20-11333R1 

Differing trabecular bone architecture in dinosaurs and mammals contribute to stiffness and limits on bone strain 

Dear Dr. Aguirre:

I'm pleased to inform you that your manuscript has been deemed suitable for publication in PLOS ONE. Congratulations! Your manuscript is now with our production department. 

Kind regards, 

on behalf of

Dr. Deepak Vashishth 

Academic Editor

PLOS ONE